# MCT1-governed pyruvate metabolism is essential for antibody class-switch recombination through H3K27 acetylation

Wenna Chi[1,2,8], Na Kang[3,4,8], Linlin Sheng [1,8], Sichen Liu[3,8], Lei Tao[1,2], Xizhi Cao[1], Ye Liu[1], Can Zhu[3], Yuming Zhang[1], Bolong Wu[1], Ruiqun Chen[1], Lili Cheng[1], Jing Wang[3], Xiaolin Sun [5,6], Xiaohui Liu[7], Haiteng Deng[7], Jinliang Yang[2], Zhanguo Li[4,5,6], Wanli Liu [3,4] ✉ & Ligong Chen [1,2] ✉

Monocarboxylate transporter 1 (MCT1) exhibits essential roles in cellular metabolism and energy supply. Although MCT1 is highly expressed in activated B cells, it is not clear how MCT1-governed monocarboxylates transportation is functionally coupled to antibody production during the glucose metabolism. Here, we report that B cell-lineage deficiency of MCT1 significantly influences the class-switch recombination (CSR), rendering impaired IgG antibody responses in *Mct1^{f/f}Mb1^{Cre}* mice after immunization. Metabolic flux reveals that glucose metabolism is significantly reprogrammed from glycolysis to oxidative phosphorylation in *Mct1*-deficient B cells upon activation. Consistently, activation-induced cytidine deaminase (AID), is severely suppressed in *Mct1*-deficient B cells due to the decreased level of pyruvate metabolite. Mechanistically, MCT1 is required to maintain the optimal concentration of pyruvate to secure the sufficient acetylation of H3K27 for the elevated transcription of AID in activated B cells. Clinically, we found that *MCT1* expression levels are significantly upregulated in systemic lupus erythematosus patients, and *Mct1* deficiency can alleviate the symptoms of bm12-induced murine lupus model. Collectively, these results demonstrate that MCT1-mediated pyruvate metabolism is required for IgG antibody CSR through an epigenetic dependent AID transcription, revealing MCT1 as a potential target for vaccine development and SLE disease treatment.

Activation of immune cells is accompanied by metabolic reprogramming to meet the increase of bioenergetic and biosynthetic requirements for fast cell growth and proliferation[1]. As the primary source of cellular energy, glucose is metabolized via glycolysis to pyruvate, which can be imported into the mitochondria for tricarboxylic acid (TCA) cycle to generate acetyl coenzyme A (Acetyl-CoA) through pyruvate dehydrogenase in the presence of sufficient oxygen, or through lactate dehydrogenase (LDH)-mediated lactate production when the oxygen

[1]School of Pharmaceutical Sciences, Key Laboratory of Bioorganic Phosphorus Chemistry and Chemical Biology (Ministry of Education), Tsinghua University, Beijing 100084, China. [2]Collaborative Innovation Center for Biotherapy, State Key Laboratory of Biotherapy and Cancer Center, West China Hospital, West China Medical School, Sichuan University, Chengdu 610065, China. [3]State Key Laboratory of Membrane Biology, School of Life Sciences, Institute for Immunology, China Ministry of Education Key Laboratory of Protein Sciences, Beijing Key Lab for Immunological Research on Chronic Diseases, Tsinghua University, Beijing 100084, China. [4]Tsinghua-Peking Center for Life Sciences, Beijing, China. [5]Department of Rheumatology and Immunology, Peking University People's Hospital, Beijing 100044, China. [6]Beijing Key Laboratory for Rheumatism Mechanism and Immune Diagnosis (BZ0135), Beijing 100044, China. [7]National Center for Protein Science, School of Life Sciences, Tsinghua University, Beijing 100084, China. [8]These authors contributed equally: Wenna Chi, Na Kang, Linlin Sheng, Sichen Liu. ✉e-mail: liulab@tsinghua.edu.cn; ligongchen@tsinghua.edu.cn

levels are sub-optimal[2]. Proliferating cells prefer to utilize aerobic glycolysis for their survival even in the presence of oxygen known as "Warburg effect", leading to generate massive lactate in the cytoplasm[3]. Then, cytosol lactate must be expelled to the extracellular micro-environment to assure the continuation of glycolysis[4]. Biochemically, lactate transport across the plasma membrane is highly dependent on monocarboxylate transporters (MCT)[5]. MCTs belong to solute carrier (SLC) transporters, which are membrane-bound proteins that play a crucial role in facilitating the transport of a diverse range of substrates[6]. SLC transporters have been reported to play crucial roles in maintaining the development, homeostasis, differentiation, and securing activation of immune cells, including T cells, NK cells, and macrophages[7]. Dysregulation of these transporters can trigger various immunologic diseases, including gout, asthma, inflammatory bowel disease, and Alzheimer's disease[8–10]. Of these transporters, monocarboxylate transporter 1 (MCT1, also known as SLC16A1) is responsible for the transportation of various monocarboxylates, including lactate and pyruvate, across the plasma membrane. As reported, MCT1 deficiency reduced cell proliferation of activated CD8$^+$ T cells and derailed their cell metabolism[11]. In addition, MCT1 is a promising target for immunosuppressive therapy as its inhibition can selectively and profoundly block the extremely rapid phase of T cell division that is crucial for an effective immune response during T cell activation[11].

Upon vaccination, B lymphocytes (B cells) with the acquired helps from other types of immune cells and cytokines are responsible for the production of pathogen antigen-specific antibodies, serving as the foundation for most of successful vaccines. These immune responses are closely correlated to the aforementioned metabolic events. In brief, mature naive B cells at quiescent state have minimal metabolic needs, but rapidly increase energy requirements and metabolic demands upon immune response initiation[1]. Indeed, there is growing evidence to show that antigen-induced B cell receptor (BCR) signaling or B-cell activating factor (BAFF)-induced BAFF-receptor signaling can strongly drive B cells for efficient glycolysis and rapid proliferation through PI3K-AKT and PKC-β signaling, leading to enhanced glucose transporter GLUT1 expression and oxidative metabolism[12]. A certain percentage of activated B cells can undergo a crucial process of class-switch recombination (CSR) mainly for the production of IgG antibodies, and in this biological event B cell-specific enzyme activation-induced cytidine deaminase (AID) is absolutely indispensable[13,14]. Currently, it is reported that various molecules including the NF-κB signaling pathway, 14-3-3 adapter protein, transcription factors, cytokines, chemokines, and histone modifications are involved in the upregulation of AID expression during CSR[15]. However, the function of MCT1 in B cells remains unclear. Specifically, it is uncertain how MCT-driven monocarboxylates transportation is functionally coupled to CSR and the subsequent antibody production, as well as the adaptive glucose metabolic dynamics of B cells during activation and differentiation upon vaccination. Moreover, the dysregulation of these biological events in B cell relevant diseases is also poorly understood.

Here, we report that B cell-specific Mct1 deficiency significantly affects IgG subtype antibody production upon immunization due to the damage of CSR and reduction in the number of germinal center B cells (GCBC). This study highlights that MCT1-mediated pyruvate metabolism is essential for CSR through an epigenetic dependent AID transcription. Furthermore, *MCT1* expression levels are found to be significantly upregulated in systemic lupus erythematosus (SLE) patients, and Mct1 deficiency can alleviate the symptoms of bm12-induced murine lupus model. These findings reveal MCT1 as a potential target for both vaccine development and SLE disease treatment.

## Results

### MCT1 transporter is required for CSR
We first isolated primary B cells from the spleen of C57BL/6J mice and have them stimulated using culture medium containing LPS and IL-4

for 2 days. We further extracted the mRNA, which was then reverse transcribed into cDNA for quantitative PCR assay. The results showed that, among all the MCT family members (*Mct1*, *Mct2*, *Mct3* and *Mct4*) for monocarboxylate transportation, the expression of *Mct1* was significantly higher than that of *Mct2*, *Mct3* and *Mct4* (Supplementary Fig. 1a). Furthermore, by combining these results to the analyses from ImmGen databases, we found that *Mct1* was highly expressed in B cell-lineage subsets including GCBC, plasmablast (PB), and plasma cell (PC) (Supplementary Fig. 1b). Furthermore, we observed that MCT1 expressed in primary B cells and 293T cells at the subcellular location of the plasma membrane (Supplementary Fig. 1c). Therefore, based on these findings, we focused our study on *Mct1* thereafter. To evaluate the function of MCT1 and monocarboxylate signaling module in B cell activation and function, we generated the B cell-lineage *Mct1*-deficient mice by breeding *Mct1*^f/f^ mice with *Mb1*^Cre^ mice according to Supplementary Fig. 2. The morphology, weight and B cell follicular patterns of the spleen did not differ between *Mct1*^f/f^*Mb1*^Cre^ mice and *Mb1*^Cre^ mice (Supplementary Fig. 3a–g). We also observed that B cell-lineage knockout (KO) of *Mct1* did not impact B cell development in the bone marrow and homeostasis in the peripheral lymphoid organs (Supplementary Fig. 4a–d). In addition, the levels of innate antibodies were comparable in these two types of mice (Supplementary Fig. 5a). All these results established the basis for us to immunize both *Mb1*^Cre^ and *Mct1*^f/f^*Mb1*^Cre^ mice with T cell-dependent (TD) model antigen NP₃₃-KLH and examine the subsequent antigen-specific antibody responses. We found that the NP-specific IgG, but not IgM, antibody production was significantly impaired in B cell-lineage *Mct1*-KO mice (Fig. 1a–c, Supplementary Fig. 5c). We also observed that the NP-specific IgG antibodies from *Mct1*^f/f^*Mb1*^Cre^ mice exhibited a weaker affinity to NP-hapten antigen compared with those from *Mb1*^Cre^ mice (Fig. 1d, Supplementary Fig. 5b). Moreover, upon the immunization with NP-Ficoll, a T cell-independent antigen, we noted that absence of *Mct1* primarily resulted in a reduced production of IgG3, but unchanged IgM antibodies (Supplementary Fig. 5d). These results indicated that *Mct1* deficiency disrupted the IgG but not IgM antibody response upon immunization.

We thus hypothesized that the damaged CSR accompanied with the alteration of GC reaction may result in a decrease of IgG antibodies. To check that, we immunized both types of mice with sheep red blood cells (SRBC) to examine GCBC formation by flow cytometry. Notably, *Mct1* deficiency resulted in a reduction in the differentiation of GCBC (Fig. 1e, f). Immunohistochemical analyses by using peanut agglutinin (PNA, a GCBC marker) further revealed a more significant reduction in the number and size of GCBC in *Mct1*^f/f^*Mb1*^Cre^ mice when compared to *Mb1*^Cre^ mice (Fig. 1g, left panel). There was also a marked reduction in Ki67-positive GCBC consistent with the lack of GCBC proliferation in GC compartment (Fig. 1g, right panel). Remarkably, the proportion of IgG1-positive but not IgM-positive B cells was significantly decreased upon *Mct1* deficiency (Fig. 1h, i). As a validation, we applied a lipopolysaccharide (LPS) and interleukin-4 (IL-4) driven in vitro B cells CSR protocol to splenic primary B cells from either *Mb1*^Cre^ or *Mct1*^f/f^*Mb1*^Cre^ mice, and consistently observed that *Mct1* deficiency resulted in a profound decrease in the percentage of IgG1-positive B cells (Fig. 1j–l). As a further validation, broad-spectrum MCTs inhibitors CHC (α-Cyano-4-hydroxycinnamic acid) and MCT1 specific inhibitor AZD3965 similarly suppressed the CSR reaction in LPS- and IL-4-stimulated primary splenic B cells in vitro (Fig. 1m–o), largely in line with the murine genetic evidence in vivo. MCT1 is a unidirectional proton-linked monocarboxylate transmembrane transporter, and its transport capacity is strictly dependent on intracellular pH. Indeed, we found that MCT1 substrates (lactate, pyruvate, acetoacetate, β-hydroxybutyrate) play different roles in regulating the intracellular pH of splenic primary B cells from either *Mb1*^Cre^ or *Mct1*^f/f^*Mb1*^Cre^ mice (Supplementary Fig. 6a–d). To examine whether the observations are due to pH changes, we manipulated the intracellular and extracellular

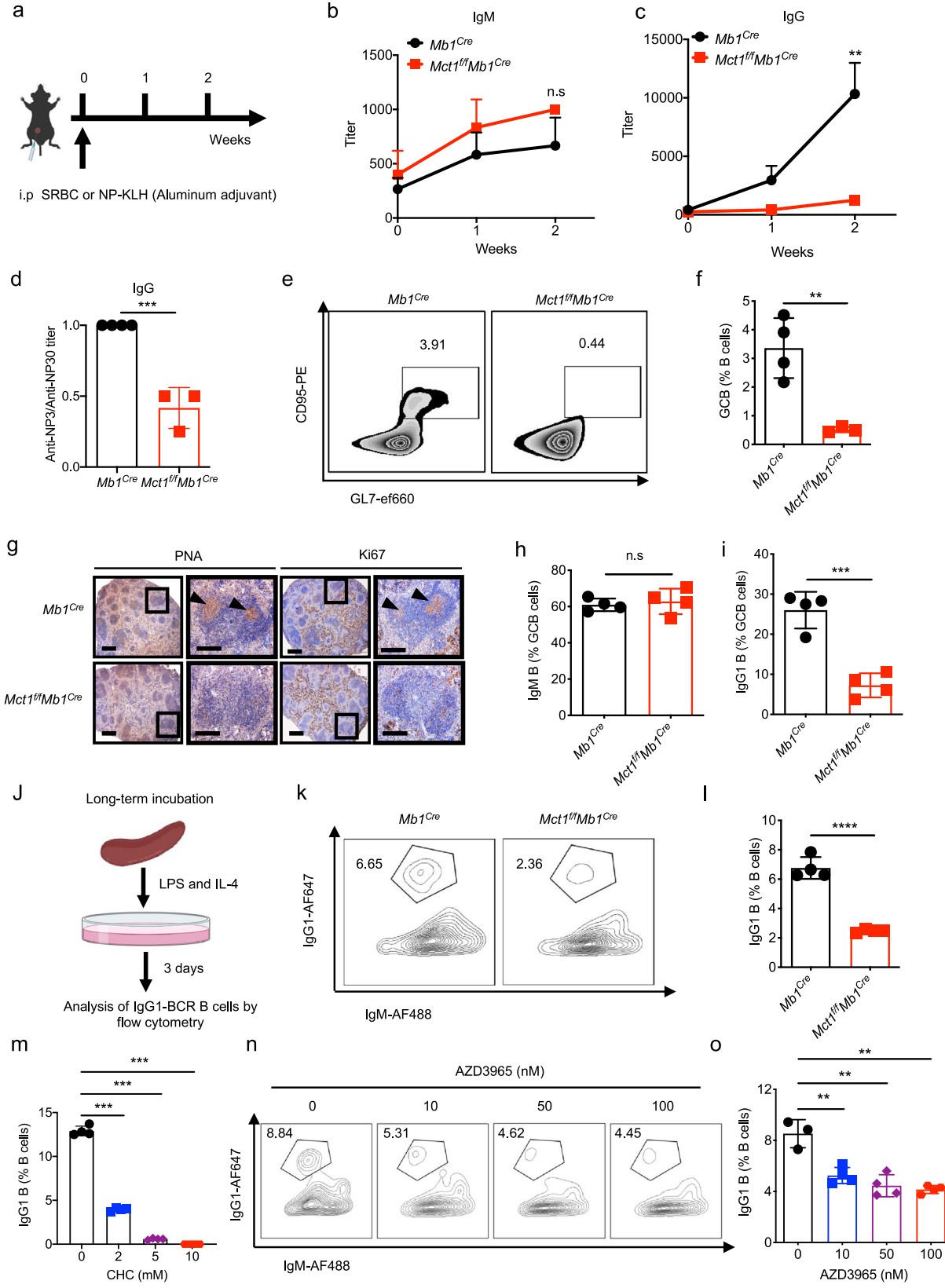

pH by treating the cells with 5-N, N-hexamethylene amiloride (HMA, an Na + /H+ antiporter inhibitor). We found that the proportion of IgG1 B cells was unchanged with or without the addition of HMA (Supplementary Fig. 6e). These results suggested that the effect of *Mct1* on CSR is independent on the intracellular pH in response to the concentration changes of MCT1 substrates.

## Glycolysis is severely inhibited in *Mct1*-deficient B cells

As the metabolic intermediates of glycolysis, both lactate and pyruvate are the substrates of MCT1 transporter[5]. To further quantify the full-phase/long-term glucose metabolism in B cells undergone activation and CSR, we stimulated these primary B cells with LPS and IL-4 for 3 days. WB analyses revealed that hexokinase 2 (HK2), aldolase B

**Fig. 1 | *Mct1* deficiency in B cells dampens the CSR. a** The schematic plot of assays in vivo. This image was adapted from Biorender. The titers of IgM (**b**), IgG (**c**) from *Mb1^{Cre}* and *Mct1^{f/f}Mb1^{Cre}* mice upon immunization (Intraperitoneal injection, i.p.) with NP$_{33}$-KLH at week 1 and 2, $n = 4$ biological replicates, $p = 0.0067$ (**c**). **d** The titers of Anti-NP3/Anti-NP30 from *Mb1^{Cre}* ($n = 4$ biological replicates, $p = 0.0004$) and *Mct1^{f/f}Mb1^{Cre}* ($n = 3$ biological replicates) mice upon immunization (i.p.) with NP$_{33}$-KLH at week 2. GCBC in *Mb1^{Cre}* and *Mct1^{f/f}Mb1^{Cre}* mice upon i.p. with SRBC at day 7 were analyzed by flow cytometry (**e**) and the percentage of GCBC is illustrated (**f**), *Mb1^{Cre}* ($n = 4$ biological replicates) and *Mct1^{f/f}Mb1^{Cre}* ($n = 3$ biological replicates), $p = 0.006$. **g** The representative immunohistochemical staining of spleens with PNA (GCBC marker) and Ki67 (proliferative cells marker) from *Mb1^{Cre}* and *Mct1^{f/f}Mb1^{Cre}* mice at day 7 with SRBC immunization (i.p.). Black arrows indicated the positive cells (brown points). Scale Bar, 250 μm and 50 μm. The percentage of IgM (**h**) and IgG1 (**i**)-BCR expressing B cells in GCBC upon immunization (i.p.) with SRBC at day 7 was analyzed by flow cytometry, $n = 4$ biological replicates, $p = 0.0005$. **j** The schematic diagram shows the experimental procedure for the analysis of IgG1 B cells after stimulation for 3 days in vitro. Flow cytometry analysis of IgM, IgG1 expressing B cells stimulated with LPS and IL-4 at day 3 from *Mb1^{Cre}* and *Mct1^{f/f}Mb1^{Cre}* mice (**k**) and the percentage of IgG1 B cells is presented (**l**), $n = 4$ biological replicates, $p < 0.0001$. **m−o** Naive B cells isolated from B6 mice were stimulated with LPS and IL-4 for 3 days with MCT1 inhibitor α-Cyano-4-hydroxycinnamic acid (CHC) or AZD3965. Cumulative data of CHC is shown in (**m**), $n = 4$ biological replicates, $p < 0.001$. Representative plots by flow cytometry and respective statistical analysis of AZD3965 are shown in (**n**, **o**), $n = 4$ biological replicates, $p < 0.01$. Data are presented as mean ± SEM of 4 mice or mean ± SD, two-way ANOVA followed by Sidak's multiple-comparisons test (**b**, **c**, **m**, **o**) or unpaired two-tailed Student's t-test (**d**, **f**, **h**, **i**, **l**), **$p < 0.01$, ***$p < 0.001$, ****$p < 0.0001$ n.s, not significant.

(ALDOB), pyruvate kinase isozymes M2 (PKM2) and lactate dehydrogenase A (LDHA) and even MCT1 were gradually upregulated, while pyruvate dehydrogenase A (PDHA) were mildly downregulated in these three days (Supplementary Fig. 7a). Additionally, lactate production was continuously increased from day 0 to day 3 in primary B cells treated with LPS and IL-4 (Supplementary Fig. 7b). These results suggest that glycolysis is highly active in B cells during the events of burst proliferation and CSR.

Therefore, we measured glucose consumption as a way to detect glycolysis efficiency, which was found to be significantly decreased in *Mct1*-deficient B cells (Fig. 2a). Additionally, *Mct1* deficiency inhibited the upregulation of HK2 expression regardless of glucose dosage (Fig. 2b). HK enzyme activity as the rate-limiting step of glycolysis, was also inhibited in *Mct1*-deficient B cells upon the stimulation by LPS and IL-4 (Fig. 2c). These results drove us to speculate that *Mct1* deficiency might induce lactate accumulation in B cells, resulting in glycolysis inhibition. However, both metabonomic analyses and general assay kit examination unexpectedly and consistently revealed that the intracellular lactate level was not increased, while its level in the supernatant was significantly decreased in *Mct1*-deficient B cells after treatment with LPS and IL-4 for 3 days (Fig. 2d, e, g, Supplementary Data S6). Importantly, in the same batch of B cells, pyruvate levels were significantly decreased in both the intracellular fractions and the extracellular supernatant (Fig. 2f, g). We also examined the other monocarboxylate substrates responsible for MCT1, including acetoacetate and β-hydroxybutyrate. We found that *Mct1* deficiency did not affect the intracellular abundance of acetoacetate and β-hydroxybutyrate (Supplementary Fig. 7c). Thus, all these results suggest that glycolysis is severely inhibited in *Mct1*-deficient B cells.

### Glucose metabolism is shifted from glycolysis to OXPHOS in *Mct1*-deficient B cells

To investigate the fate of lactate and pyruvate in *Mct1*-deficient B cells, we employed the glucose isotopomer tracing ($^{13}$C-isotope-labeled glucose) to examine the glucose metabolic dynamics of B cells undergoing CSR (Fig. 3a, b). Primary B cells were negatively sorted from either *Mb1^{Cre}* or *Mct1^{f/f}Mb1^{Cre}* mice, and stimulated with medium containing LPS and IL-4 on day 0. On day 2, the culture medium was replaced with U-$^{13}$C$_6$ glucose, and both cells and medium were harvested and detected at 30, 60 and 180 min after replacement. Interestingly, the tracing experiments seemed to indicate that OXPHOS was enhanced in *Mct1*-deficient B cells. In comparison to *Mb1^{Cre}* B cells, we observed a more obvious accumulation of the intracellular glucose (m + 6) in *Mct1*-deficient B cells at all three time points (30, 60 and 180 min), and only at 180 min time point for that from the supernatant fraction. However, both lactate (m + 3) and pyruvate (m + 3) were reduced in the supernatant and accumulated in the intracellular space at all three time points (30, 60 and 180 min). Pyruvate and lactate were transitory accumulated in the intracellular fraction of *Mct1*-deficient B cells (Fig. 3c, d). Remarkably, doubly (m + 2) citrate, succinate and malate were drastically increased in the intracellular fraction of *Mct1*-deficient B cells at all three time points (30, 60 and 180 min) (Fig. 3e). These analyses, combined with the changes of intracellular lactate and pyruvate at day 3, consistently suggested that accumulated lactate and pyruvate might be redirected into the TCA cycle for OXPHOS.

It is worth noting that the basal oxygen consumption rate (OCR) was increased in *Mct1*-deficient B cells treated with LPS and IL-4 for 2 days (Fig. 3f), resulting in a similar level of ATP (Fig. 3g). We thus examined cell proliferation and apoptosis by propidium iodide (PI) and Ki67, respectively, in primary splenic B cells upon stimulation with LPS and IL-4. The B cells from these two types of mice showed similar proliferation and survival rates at both day 0 and day 3, with comparable cell sizes (Supplementary Fig. 8a–c), suggesting that the *Mct1* transporter did not affect B cell survival. Mitochondrial fusion is known to favor OXPHOS[16]. We thus used electron microscopy (EM) to examine mitochondria changes in these two types of B cells upon the activation by LPS and IL-4 for 3 days. We found that B cells from *Mb1^{Cre}* mice exhibited small, round and prominent mitochondria in the cytoplasm, whereas *Mct1*-deficient B cells distributed tubular mitochondria (Fig. 3h). In addition, *Mct1* deficiency increased the mitochondrial potentials, while the mitochondrial mass was unchanged (Supplementary Fig. 9a, b). To further explore the differences in mitochondrial morphology in these two types of B cells, we examined B cell mitochondria by high sensitivity structured illumination microscope (HIS-SIM). We found that *Mct1*-deficient B cells contained elongated mitochondria, as aforementioned in EM analyses (Supplementary Fig. 9c, d). Consistent with mitochondrial fusion, membrane fusion mediators *Mfn1*, *Mfn2* and *Opa1*, were all upregulated in *Mct1*-deficient B cells at both mRNA and protein levels (Fig. 3i, Supplementary Fig. 9e). A growing body of evidence has showed that deacetylation of MFN1 activates mitochondrial fusion[17,18]. Indeed, we observed that the level of MFN1 acetylation was consistently low in *Mct1*-deficient B cells (Supplementary Fig. 9f). Taken together, these results suggest that *Mct1* deficiency initially causes an excessive accumulation of lactate within the cytosol of B cell, which then inhibits glycolysis and prompts the conversion of lactate back into pyruvate. Then, these pyruvates are redirected towards mitochondrial pyruvate oxidation, ultimately leading to a shift in glycolytic carbon flux towards mitochondrial pyruvate oxidation. This redirection of metabolic pathways allows *Mct1*-deficient B cells to maintain energy homeostasis, which is essential for their survival and function (Fig. 3j).

### Pyruvate is functionally coupled with the MCT1 transporter to regulate the expression of AID

AID plays a critical role in CSR. A previous study showed that CSR was significantly suppressed in *Aicda*-deficient mice and in patients with *AICDA* loss-of-function mutations (autosomal recessive hyper IgM syndrome type 2, HIGM2)[19]. Global transcriptome analysis of primary B cells from both *Mb1^{Cre}* and *Mct1^{f/f}Mb1^{Cre}* after the treatment with LPS and IL-4 for 3 days revealed that *Aicda* expression

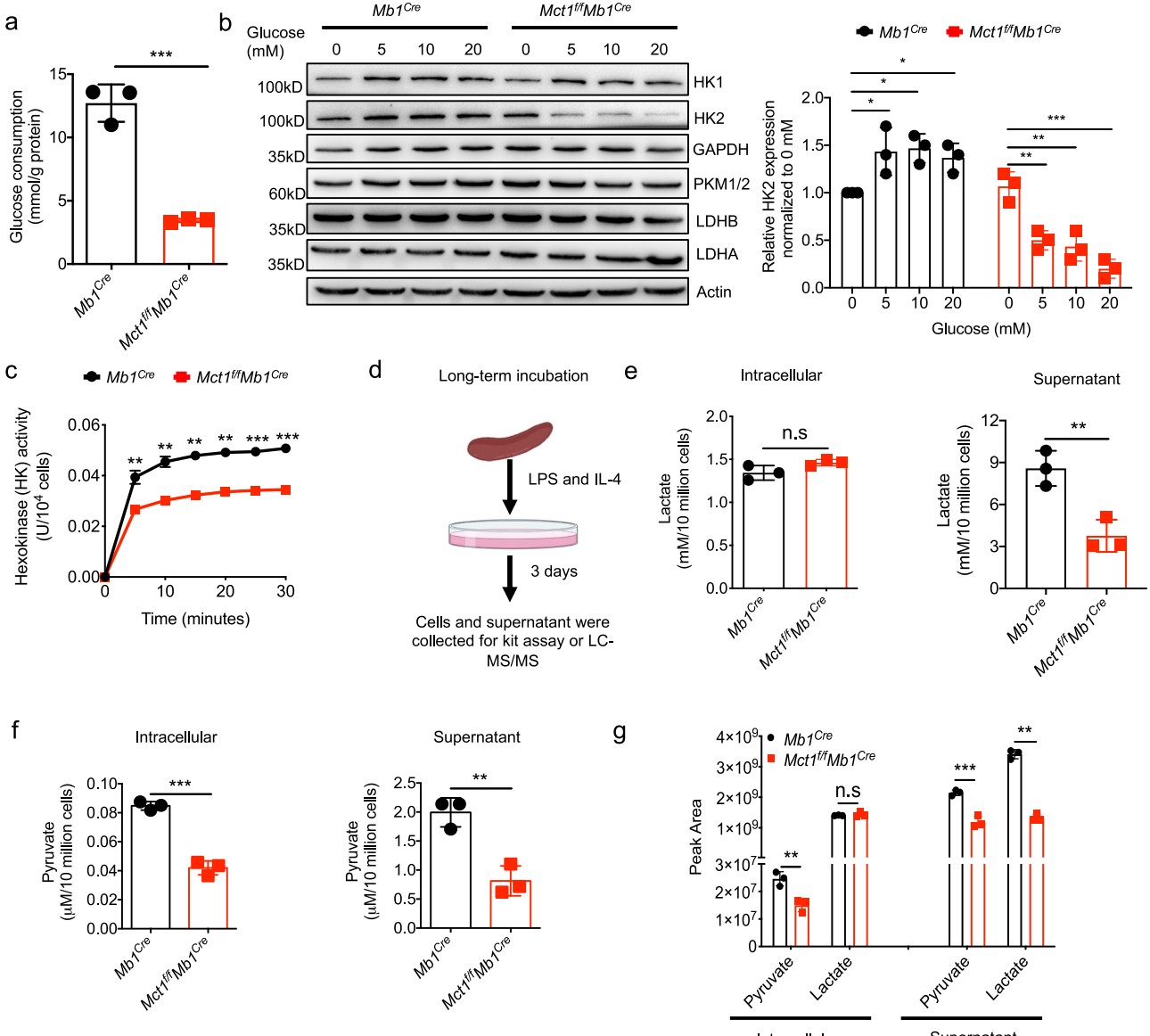

**Fig. 2 | *Mct1* deficiency inhibits the glycolysis metabolism of B cells. a** The naive B cells isolated from the *Mb1*^Cre and *Mct1*^f/f*Mb1*^Cre cells were stimulated with LPS and IL-4 for 3 days. The representative glucose consumption is shown, *n* = 3 biological replicates, *p* = 0.5546. **b** Naive B cells isolated from the *Mb1*^Cre and *Mct1*^f/f*Mb1*^Cre mice were cultured with different levels of glucose and stimulated with LPS and IL-4 for 3 days. The expression of enzymes of glucose metabolism was presented by immunoblot (left) and its quantification (right), *n* = 3 biological replicates, *p < 0.05, **p < 0.01, ***p < 0.001. **c** Hexokinase activity of B cells isolated from the *Mb1*^Cre and *Mct1*^f/f*Mb1*^Cre mice treated with LPS and IL-4 for 3 days was measured using a hexokinase activity assay kit, *n* = 3 biological replicates, **p < 0.01, ***p < 0.001. **d** The schematic diagram shows the experimental procedure for metabolite analysis from cells or supernatants after stimulation for 3 days in vitro. This image was adapted

from Biorender. **e–g** Naive B cells isolated from *Mb1*^Cre and *Mct1*^f/f*Mb1*^Cre mice were stimulated with LPS and IL-4 for 3 days, *n* = 3 biological replicates, **p < 0.01, ***p < 0.001. The concentration of lactate in intracellular (left) and supernatant (right) was determined by lactate assay kit (**e**). The relative level of lactate in intracellular (left) and supernatant (right) was determined by metabonomics (**g**). The concentration of pyruvate in intracellular (left) and supernatant (right) was determined by pyruvate assay kit (**f**). The relative level of pyruvate in intracellular (left) and supernatant (right) was determined by metabonomics (**g**). Data are presented as mean ± SD, two-way ANOVA followed by Sidak's multiple-comparisons test (**c**) or unpaired two-tailed Student's t-test (**a**, **b**, **e–g**), *p < 0.05, **p < 0.01, ***p < 0.001, n.s, not significant. Source data are provided as a Source Data file.

was significantly reduced in *Mct1*-deficient B cells (Fig. 4a). MCT1- and AID-specific quantitative RT-PCR and WB experiments also confirmed that *Mct1* deficiency induced a drastic downregulation of AID at both mRNA and protein levels (Fig. 4b, c). Based on the above evidence, we hypothesized that the pyruvate and lactate, which act as substrates for the MCT1 transporter, may play reciprocal roles in ensuring the efficiency of CSR. To explicitly evaluate the effects of lactate and pyruvate, we supplemented exogenous pyruvate or lactate to B cells from the WT control *Mb1*^Cre mice (WT control B cells) treated with LPS and IL-4 from day 1 to day 3. We

observed that the total number of cells was unchanged when increasing sodium pyruvate concentrations, while the proportion of IgG1 B cells was enhanced (Fig. 4d, Supplementary Fig. 10a). However, the proportion of IgG1 primary B cells was unchanged with the addition of exogenous lactate in WT control B cells (Supplementary Fig. 10b). These data suggested that the percentage of IgG1+ B cells was increased with the addition of pyruvate. Moreover, in the absence of glucose, AID levels were increased upon the addition of pyruvate in WT control B cells from *Mb1*^Cre mice (Fig. 4e).

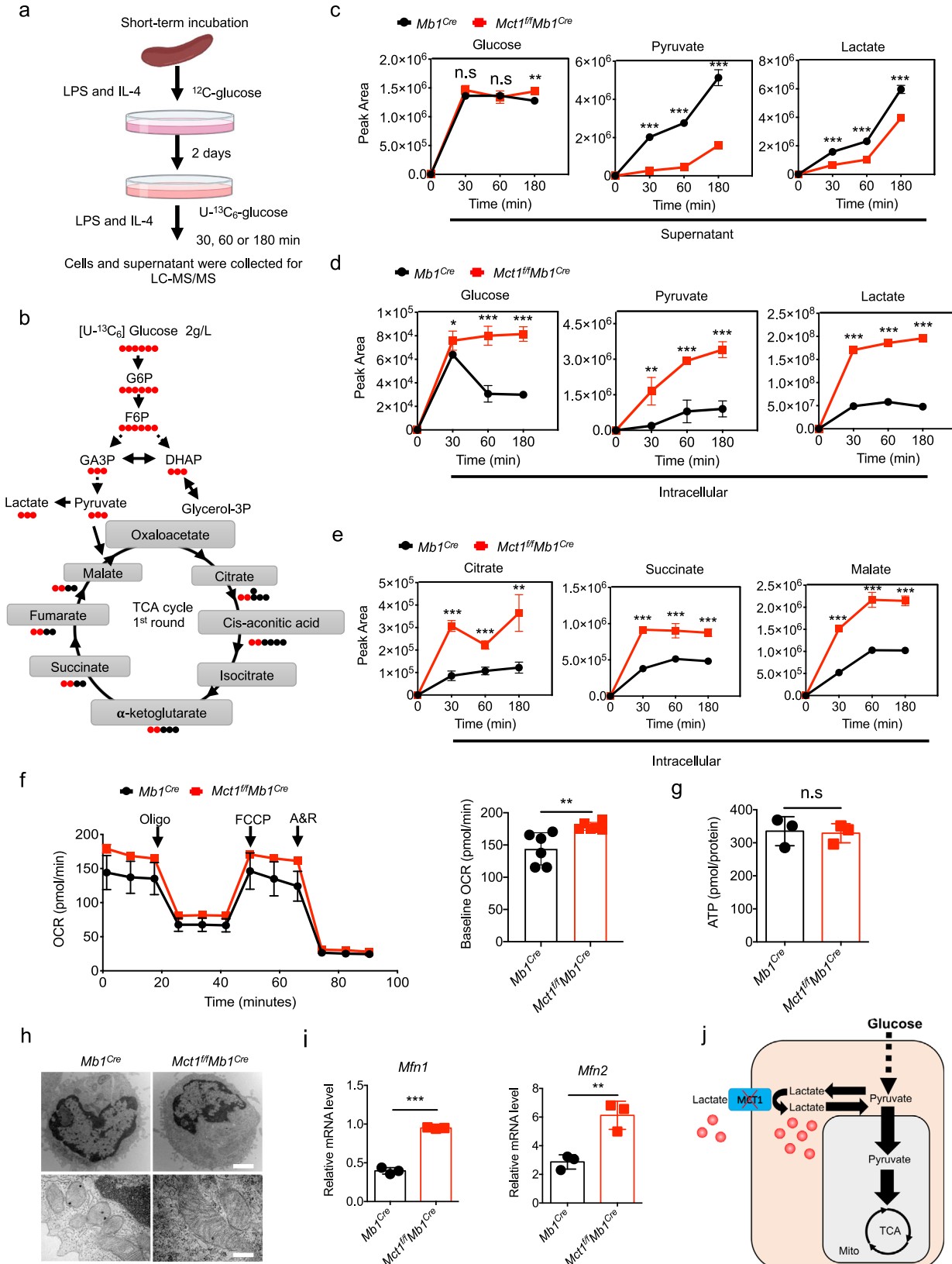

In marked contrast, exogenous pyruvate failed to upregulate the proportion of IgG1 B cells in the condition of *Mct1* deficiency (Fig. 4f). To assess whether pyruvate was sufficient to exert these intensive effects once reaching the cytosol, we circumvented the requirement of the MCT1 for pyruvate transportation by supplying the *Mct1* deficiency B cells with methyl-pyruvate (MePyr), a pyruvate analog that can

penetrate the plasma membrane independent on MCT1 due to its lipophilic feature. Indeed, the proportion of IgG1 B cells was increased by MePyr supplementation (Fig. 4g). In addition, the expression of AID was strengthened by MePyr treatment in the absence of glucose in *Mct1*-deficient B cells (Fig. 4h). To further validate the effect of pyruvate on CSR, we established a *Mct1*-deficient A20 cell line (A20-

**Fig. 3 | *Mct1* deficiency promotes the glucose flux into the mitochondrial pyruvate oxidation. a** The schematic diagram shows the experimental procedure for metabolite analysis from cells or supernatants after stimulation with LPS and IL-4 for 2 days, replacing with U-$^{13}C_6$ glucose tracing 30 min, 60 min and 180 min to do LC-MS/MS in vitro. This image was adapted from Biorender. **b** The schematic plot of $^{13}C_6$ glucose tracing. **c–e** Naive B cells isolated from the *Mb1$^{Cre}$* and *Mct1$^{f/f}$Mb1$^{Cre}$* mice were stimulated with LPS and IL-4 for 2 days. Cells were replenished with U-$^{13}C_6$ glucose medium, which was harvested for glucose flux after 30, 60 and 180 min. The glucose (m + 6), pyruvate (m + 3) and lactate (m + 3) levels were checked in the supernatant (**c**) and intracellular (**d**), $n = 4$ biological replicates, *$p < 0.05$, **$p < 0.01$, ***$p < 0.001$. The intracellular levels of citrate (m + 2), succinate (m + 2) and malate (m + 2) (**e**), $n = 4$ biological replicates, **$p < 0.01$, ***$p < 0.001$. **f** Naive B cells isolated from the *Mb1$^{Cre}$* and *Mct1$^{f/f}$Mb1$^{Cre}$* mice were stimulated with LPS and IL-4 for 2 days.

Oxygen consumption rate (OCR) was determined by Seahorse with six biological replicates ($n = 6$ biological replicates, $p = 0.0076$). Baseline OCR (right) was calculated according to OCR (left). Oligo: Oligomycin, 2 μM; FCCP, 1 μM; A: Antimycin A, 1 μM; R: rotenone, 1 μM. **g** B cells from *Mb1$^{Cre}$* and *Mct1$^{f/f}$Mb1$^{Cre}$* mice with LPS and IL-4-treated for 3 days were harvested for ATP detection by a ATP assay kit with triplicate ($n = 3$ biological replicates). **h** EM analysis of mitochondrial morphology in B cells with LPS/IL-4 treatment for 2 days. Scale Bar, 2 μm (upper) and 200 nm (lower). **i** RT-PCR analysis of *Mfn1* and *Mfn2* in B cells after LPS/IL-4 activation for 2 days, $n = 3$ biological replicates, $p = 0.0004$(left), $p = 0.007$(right). **j** Schematic depicting rewiring of the metabolism in *Mct1*-deficient B cells. Data are presented as mean ± SD, two-way ANOVA followed by Sidak's multiple-comparisons test (**c–e**) or unpaired two-tailed t-test (**f, g, i**), *$p < 0.05$, **$p < 0.01$, ***$p < 0.001$, n.s not significant.

sg*Mct1*) using the CRISPR-Cas9 technique (Fig. 4i). Consistent with the results from primary B cells, in the absence of glucose, the expression of AID was upregulated by pyruvate in a dose-dependent manner in WT-control A20 cells, and by MePyr in A20-sg*Mct1* cells, respectively (Fig. 4j–k). However, the levels of AID were unaltered by lactate treatment in both WT control and A20-sg*Mct1* cells (Supplementary Fig. 10c). Taken together, the above results indicate that pyruvate but not lactate is a potent factor securing the expression of AID during the CSR in B cells.

### MCT1 maintains chromatin accessibility at the promoter region of *Aicda* gene in CSR B cells

To investigate the mechanism of action during MCT1-mediated CSR, we performed RNA sequencing again to examine the global transcription dynamics in primary B cells from either *Mb1$^{Cre}$* or *Mct1$^{f/f}$Mb1$^{Cre}$* mice before and after the activation by LPS and IL-4 for 3 days. The analyses revealed that there are no major differences for these two types of B cells on day 0, however on day 3 a total of 453 transcripts were differentially expressed, among which 285 transcripts, including *Aicda*, were downregulated in *Mct1*-deficient B cells (Supplementary Fig. 11a, Supplementary Data S3). The results for the transcription of *Aicda* is consistent with the snapshot RNA sequencing results for these two types of B cells as aforementioned in Fig. 4a, while more importantly this new dynamic analysis for global transcription revealed that the genetic transcriptions of the protein molecules responsible for histone acetylation were also highly disturbed (Supplementary Fig. 11b). Based on these results, we speculate that *Mct1* deficiency may impair the upregulated transcription of *Aicda* upon that is required for B cell CSR potentially through a model involving in histone acetylation. To explore the biochemical mechanism, we focused on chromatin accessibility as it is widely known that global histone 3 (H3) acetylation modifications at lysine residues including K9, K14, K18 and K27 are known to be associated with chromatin opening and transcription. Indeed, H3K9, H3K14 and H3K27 acetylation were all reduced in *Mct1*-deficient B cells, whereas H3K18 acetylation and H3 tri-methylation at K27 were largely unaltered (Fig. 5a). Since H3K27 is the most severely affected, we performed further analyses and showed that H3K27 acetylation may be involved in the regulation through the histone acetyltransferases (CBP or P300) and histone deacetylases (HDAC or SIRT) (Fig. 5b), consistent with the literature reports regarding the function of these biochemical mechinaries[20].

To further investigate MCT1-mediated CSR through H3K27 acetylation, we performed transposase-accessible chromatin by high-throughput sequencing (ATAC-seq) and genome-wide analysis of H3K27 acetylation by chromatin immunoprecipitation sequencing (ChIP-seq) to evaluate the global gene expression. We observed that a total of 221 H3K27 acetylation loci (scaled to 2 kb upstream and downstream of TSS with an extension of 2 kb) were significantly different in the primary B cells from *Mb1$^{Cre}$* and *Mct1$^{f/f}$Mb1$^{Cre}$* mice upon treatment with LPS and IL-4 for 3 days, of which 135 were decreased in *Mct1*-deficient B cells (Supplementary Fig. 11b).

Differentially acetylated loci were mostly mapped to the promoter regions (Supplementary Fig. 11c). Interestingly, there were no obvious differences in H3K27 acetylation within the loci of housekeeping gene *Actb*, or the key genes of GC formation, including *Bcl6, Irf4, Foxo1, Bach2, Myc, Mef2b* and *Egr3* (Supplementary Fig. 12a). Of note, OBF1 and Oct molecules are known to participate in GC transcriptional program[21], but in our case of *Mct1* deficiency, there were no obvious differences in H3K27 acetylation within these two loci (Supplementary Fig. 12b). Remarkably, *Mct1* deficiency led to a significantly reduced accessibility of the *Aicda* locus (Fig. 5c). We further analyzed H3K27Ac recruitment to the *Aicda* locus by ChIP-qPCR assays, which showed a significantly decreased abundance of *Aicda* promoter in *Mct1*-deficient B cells treated with LPS and IL-4 for 3 days (Fig. 5d). Taken together, these results suggest that the MCT1 transporter may promote the expression of AID by ensuring the acetylation of H3K27.

### Pyruvate promotes the acetylation of histone H3K27 in B cells

As aforementioned in Fig. 5b, we measured the expression of CBP, HDAC and SIRT1 in B cells treated with LPS and IL-4 for 3 days by western blotting. HDAC2 was upregulated in *Mct1*-deficient B cells in comparison to control B cells, and we consistently observed a decrease of CBP and an increase of SIRT1 in *Mct1*-deficient B cells (Fig. 5b). As a validation of this observation, we found that NAM, a SIRT inhibitor, rescued the decreased proportion of IgG B cells and AID expression in *Mct1*-deficient B cells (Fig. 5e, f), confirming again that the histone acetylation indeed plays an essential role in B cell CSR. Furthermore, quantitative proteomics analysis of a diverse set of acetylated sites revealed a consistent 70-80% incorporation of $^{13}C$-labeled pyruvate at the H3K27Ac site after U-$^{13}C_3$-pyruvate treatment of A20 cells for 25 h (Fig. 5g), suggesting that pyruvate is involved in the acetylation of histone H3K27. To evaluate the contribution of pyruvate, we cultured B cells from *Mb1$^{Cre}$* mice in pyruvate-containing media with LPS and IL-4 from 1 to 3 days, which indicated that H3K27Ac levels were significantly upregulated by the addition of the pyruvic acid (Fig. 5h). In contrast, exogenous pyruvate failed to increase the levels of H3K27Ac levels in *Mct1*-deficient B cells, while the levels of H3K27Ac were rescued only by the supplementation of the MePyr (Fig. 5i), in line with the aforementioned results. In addition, we examined the mitochondrial and cytosolic concentrations of Acetyl-CoA, and found that Acetyl-CoA was increased in both mitochondria and cytoplasm in *Mct1*-deficient B cells, especially in the mitochondria (Supplementary Fig. 12c). These data suggest that mitochondrial Acetyl-CoA may primarily support TCA cycle activity rather than histone acetylation in B cells. Collectively, these findings demonstrate that the MCT1 transporter maintains a high intracellular concentration of pyruvate for promoting sufficient histone acetylation of H3K27 to ensure optimal *Aicda* transcription.

### MCT1 is highly expressed in B cells from SLE patients and serves as a potential therapeutic target for SLE treatment

SLE, an autoimmune disease, is characterized by a loss of self-tolerance leading to abnormal expansion of hyper-activated B cells[22,23].

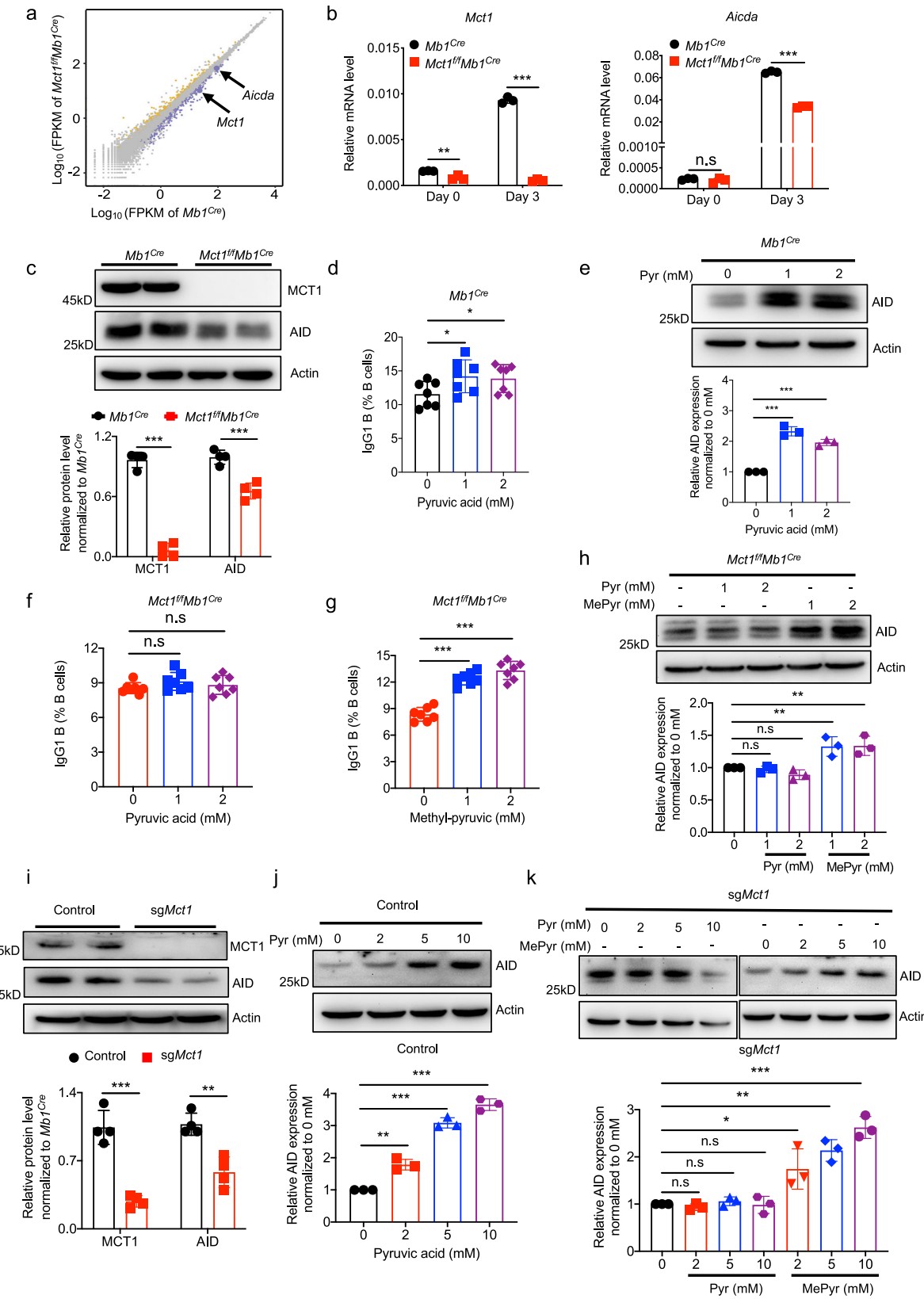

Autoreactive B cells in SLE patients undergo B cell activation, CSR, and differentiate into plasma cells that secrete abundant pathogenic autoantibodies, mainly IgG subclasses[24,25]. Given the essential function of MCT1 in securing B cell CSR, we investigated the role of the MCT1 transporter in SLE. We first analyzed a GEO database (GSE118254) containing different B cell subsets from either SLE patient or criteria-matched healthy controls (HCs), and found that the expression of *MCT1* was evaluated in the IgG class-switched memory B cells and activated naive B cells from SLE patients in comparison to HCs (Fig. 6a). In addition, two additional GEO databases (GSE10325 and GSE4588) sequencing primary B cells with mixed subsets also showed that *MCT1* was highly expressed in B cells from SLE patients than those

**Fig. 4 | Pyruvate supports the CSR via securing the expression of AID.**
**a** Scatterplot of gene expression in B cells from $Mb1^{Cre}$ and $Mct1^{f/f}Mb1^{Cre}$ mice stimulated with LPS and IL-4 for 3 days, as determined by RNA-seq. Yellow: upregulated genes. Blue: downregulated genes. **b** Naive B cells isolated from $Mb1^{Cre}$ and $Mct1^{f/f}Mb1^{Cre}$ mice were stimulated with LPS and IL-4 for 3 days. mRNA expression of $Mct1$ and $Aicda$ was analyzed by qPCR, $n = 3$ biological replicates, $**p < 0.01$, $***p < 0.001$. **c** As above conditions, analysis protein level of MCT1 and AID by immunoblot, $n = 4$ biological replicates, $p < 0.001$. **d, e** The B cells from WT control $Mb1^{Cre}$ mice were treated with LPS and IL-4 for 3 days and cultured in the presence or absence of the Pyr (Sodium pyruvate, 1 mM, 2 mM). The proportion of IgG1 B cells caused by Pyr was analyzed by flow cytometry (**d** $n = 7$ biological replicates, $p < 0.05$). The expression of AID caused by Pyr was assessed by immunoblot (**e** $n = 3$ biological replicates, $p < 0.001$). **f–h** $Mct1$-deficient B cells were pretreated with Pyr (**f** $n = 7$ biological replicates) or methyl-pyruvate (MePyr, **g**, $n = 7$ biological replicates, $p < 0.001$) co-stimulated by LPS and IL-4 for 3 days. Protein abundance of AID was analyzed by western blotting (**h**, $n = 3$ biological replicates, $p < 0.01$). **i** Western blotting bands and their quantification analysis of MCT1 and AID expression of control and sg$Mct1$ A20 cells, $n = 4$ biological replicates, $p = 0.000346$(left), $p = 0.002247$(right). **j, k.** The control A20 cells were cultured with various concentrations of Pyr (**j**) for 24 h, $n = 3$ biological replicates, $**p < 0.01$, $***p < 0.001$. The sg$Mct1$ A20 cells were cultured with various concentrations of Pyr or MePyr (**k**) for 24 h, $n = 3$ biological replicates, $*p < 0.05$, $**p < 0.01$, $***p < 0.001$. Data are presented as mean ± SD, two-way ANOVA followed by Sidak's multiple-comparisons test (**b**) or two-tailed unpaired t-test (**c–k**), $*p < 0.05$, $**p < 0.01$, $***p < 0.001$, n.s, not significant. Source data are provided as a Source Data file.

from HCs (Fig. 6b). As a validation for these analyses from the public GSE database, we established a cohort containing both SLE patients and HCs, and found that even the PBMCs from the SLE patients also exhibited significantly increased expression of $MCT1$, compared to those from HCs at mRNA level (Fig. 6c). Moreover, we did not find the upregulation of $MCT1$ in three other types of autoimmune diseases including RA, MS and SS (Supplementary Figs. 13a–13c). To validate these clinical cohort analyses, we used a bm12-induced murine lupus model that was initiated by injecting either $Mb1^{Cre}$ or $Mct1^{f/f}Mb1^{Cre}$ mice with 10 million splenocytes from age- and gender-matched bm12 mice (Fig. 6d). The $Mb1^{Cre}$ mice showed a larger spleen than $Mct1^{f/f}Mb1^{Cre}$ mice upon the induction (Fig. 6e). Correspondingly, the proportion of both GCBC and IgG1 B cells were reduced (Fig. 6f, g). Importantly, the serum IgG antibodies against dsDNA were significantly decreased in $Mct1^{f/f}Mb1^{Cre}$ mice (Fig. 6h). In addition, $Mct1$ deficiency alleviated the nephritis in $Mct1^{f/f}Mb1^{Cre}$ mice, likely due to the reduced accumulation of IgG autoantibodies in the glomerulus (Fig. 6i, j). Additionally, a SLE therapeutic agent exhibited some certain levels of capability to decrease $MCT1$ expression in vivo during treatment of lupus nephritis patients (Supplementary Fig. 13d). Notably, effect of clinically approved SLE drug, dexamethasone, was particularly significant in decreasing the expression of $MCT1$ in PBMCs in vitro (Supplementary Fig. 13e). As a further validation, we also tried to treat bm12-induced SLE mice with a MCT1 inhibitor, CHC, and found that anti-dsDNA was significantly decreased (Fig. 6k, l, Supplementary Fig. 13f). All of these results confirm a role for MCT1 in the potential pathogenesis and therapies of SLE.

## Discussion

Naive B cells are typically characterized by their low metabolic demands. Recently, a study revealed that activated B cells, rather than GCBC, generated higher levels of phosphorylated glucose and lactate during CSR[26]. Therefore, our study focuses on activated B cells during CSR, proposing that ablation of MCT1 shifts glucose metabolism from glycolysis to OXPHOS, reducing AID expression by regulating H3K27 acetylation, which, in turn, hinders CSR and participates in SLE.

Our study showed that glycolysis was inhibited due to HK2 downregulation and lactate accumulation upon the deficiency of $Mct1$. OXPHOS is enhanced with the possible purpose to maintain cellular energy demand. These results suggest that metabolic remodeling occurs in B cells by $Mct1$ deficiency. We also observed that pyruvate was significantly reduced along with impaired CSR and decreased AID under $Mct1$ deficiency, which can be restored by exogenous methyl-pyruvate supplementation, indicating that a reduced level of pyruvate is responsible for the declined expression of AID. A previous study indicated that acetylation of histone H3K27 plays a role in chromatin opening in B cells by histone modify enzymes, including CBP, SIRT1 and HDAC[27,28]. Consistent with these studies, we also found that the histone H3K27 acetylation was inhibited, thereby curtailing the $Aicda$ transcription in $Mct1$ deficient B cells. Previously, it has been reported that short-chain fatty acids play a critical role in regulating AID and

Blimp expression, and thus determine somatic hypermutation and plasma cell differentiation[29]. However, in our in vitro experiments, we did not add any fatty acids to the medium, and no short-chain fatty acids were detected in the metabolome.

In our study, Mct1 deficiency decreased the number of GCBC and AID expression. Initially, it seems contrary to our results that AID-/- mice show enhanced GC reactions and a hyper IgM phenotype[30,31]. However, it is known that the phenotype of global or conditional genetic knockout mice for a certain enzyme may not always perfectly match to the genetically modified mice expressing enzyme activity compromised version or to mice expressing decreased levels of an enzyme (knock-down version). In the case of AID enzyme, it has been reported that mice expressing AID with impaired catalytic function exhibit a lower percentage of GCBC, lower IgG/IgM production compared to wild-type mice[32]. Vhl loss-of-function decreased Aicda mRNA expression along with the decrease in IgM and IgG antibodies, but GCBC were decreased, which is completely consistent with our results[33]. Thus, AID enzyme activity compromised mice or mice expressing decreased AID levels may differ from AID deficient mice to exhibit the exactly same phenotype, namely enhanced GC reactions and a hyper IgM phenotype, although the essential phenotype of a dysfunction of GC response is largely consistent. In addition, it's also well documented that CSR is triggered prior to differentiation into GCBC and is greatly diminished in GCs[34]. Therefore, we propose that $Mct1$ deficiency leads to AID-associated CSR blockade, which then leads to a reduction in the number of GCBC, both of which together explain the reduced antibody production observed in these mice.

While Acetyl-CoA is one of the substrates for histone acetylation, the level of histone acetylation is regulated and influenced by various factors, including the activity of histone acetyltransferase (HAT) and histone deacetylase (HDAC), the availability of histone substrates, and the expression level of histone-modifying enzymes, among others[35,36]. Acetyl-CoA has a wide range of functions in living organisms, including energy metabolism, fatty acid metabolism, and cholesterol synthesis[37,38]. Furthermore, direct evidence indicates that, although ACLY knockout cell lines elevated intracellular Acetyl-CoA levels caused by the surrogate effect of ACSS2, histone modification levels are downregulated in ACLY knockout cell lines compared to wild-type cells[39]. There are also studies showing that the reduced citrate levels in cells due to mitochondrial damage by overexpression of Bcl-xL resulted in a decrease of the levels of Acetyl-CoA and N-α-acetylated proteins in the cytoplasm, which did not affect histone acetylation[40]. Therefore, although elevated levels of Acetyl-CoA may contribute to histone acetylation, it is not a definitive indication of elevated histone acetylation. Mechanistically, it is known that Acetyl-CoA in the nucleus is responsible for histone acetylation. Currently there are two main ways for Acetyl-CoA to enter the nucleus[41]. One way is that pyruvate enters the tricarboxylic acid cycle to produce Acetyl-CoA. Subsequently, these Acetyl-CoA are transported into the nucleus. The second way is that pyruvate can directly produce Acetyl-CoA in the nucleus in the

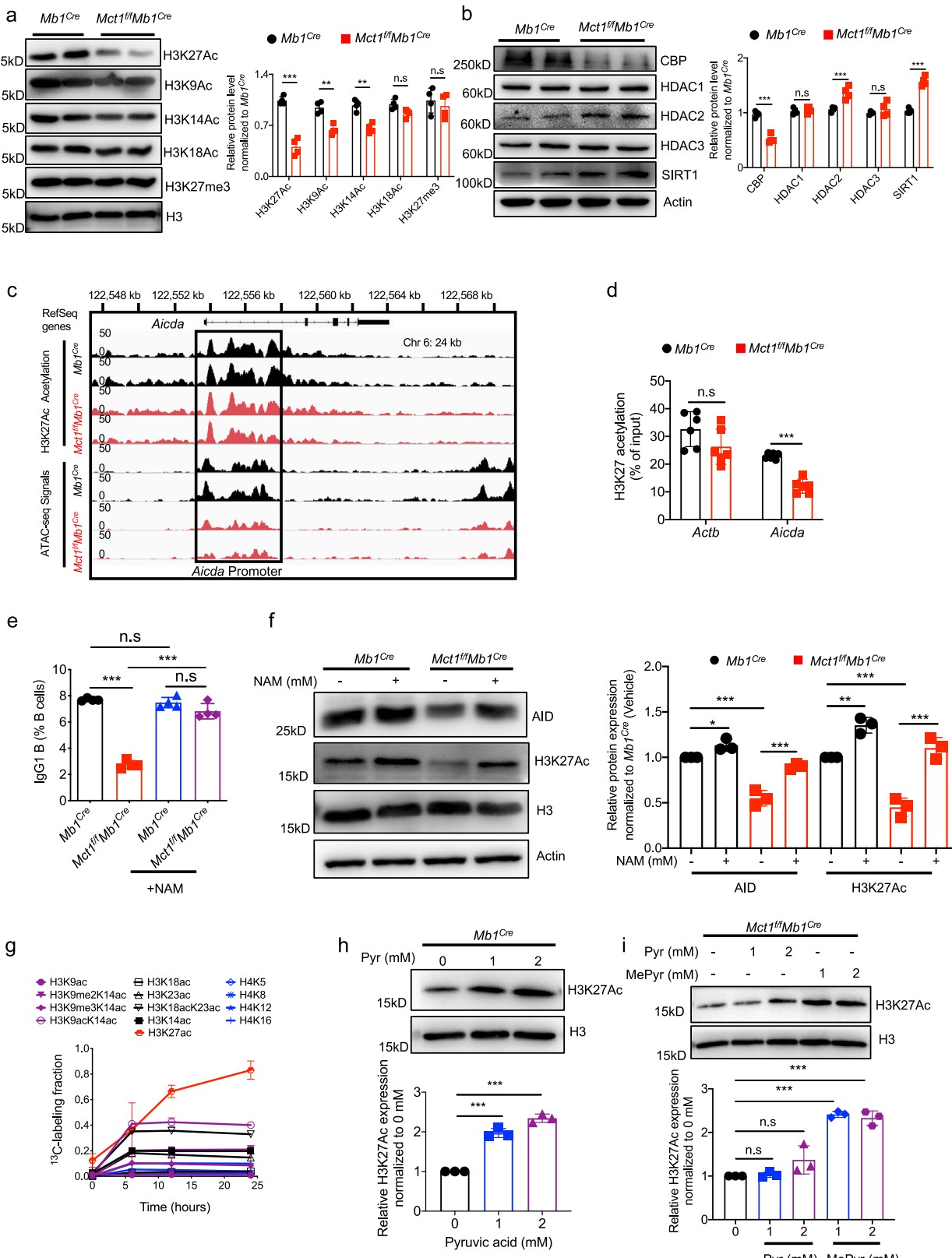

presence of pyruvate dehydrogenase complex (PDC)[42]. However, with current technologies, it is very difficult to separate the nuclear and mitochondria pools of Acetyl-CoA in mouse B cells. Thus, our results can only suggest that the elevated Acetyl-CoA in the whole cells or in the cytosol may be due to the drastically increased content of Acetyl-CoA in mitochondria. Furthermore, quantitative proteomics analysis of a diverse set of acetylated sites revealed a

consistent 70-80% incorporation of [13]C-labeled pyruvate at the H3K27Ac site after U-[13]C$_3$-pyruvate treatment. To evaluate the contribution of pyruvate, we cultured B cells from Mb1[Cre] mice in media containing pyruvate. Our results indicated that H3K27Ac levels were significantly upregulated by the addition of pyruvate (Fig. 5g–i). Therefore, these results suggest that pyruvate can directly affect histone acetylation independently of Acetyl-CoA.

**Fig. 5 | MCT1 supports CSR by glucose metabolism, which provides substrates for the histone acetylation. a** Western blotting bands and their quantification analysis of H3K27Ac, H3K9Ac, H3K14Ac, H3K18Ac and H3K27me3 proteins in B cells from *Mb1^Cre* and *Mct1^{f/f}Mb1^Cre* mice stimulated with LPS and IL-4 for 3 days, $n = 4$ biological replicates, **$p < 0.01$, ***$p < 0.001$. **b** Western blotting bands and their quantification analysis of CBP, HDAC and SIRT in B cells treated with LPS and IL-4 for 3 days, $n = 4$ biological replicates, $p < 0.001$. **c** Analysis of H3K27 acetylation at the *Aicda* gene promoter with ChIP-seq and ATAC-seq in B cells from *Mb1^Cre* and *Mct1^{f/f}Mb1^Cre* mice stimulated with LPS and IL-4 for 3 days, $n = 3$ biological replicates. **d** Chromatin accessibility at the *Aicda* promoter analyzed by the Chip-qPCR in B cells from *Mb1^Cre* and *Mct1^{f/f}Mb1^Cre* mice stimulated with LPS and IL-4 for 3 days ($n = 6$ biological replicates). **e, f** Naive B cells isolated from *Mb1^Cre* and *Mct1^{f/f}Mb1^Cre* mice were co-stimulated with LPS and IL-4 and added NAM (Nicotinamide, sirtuins

inhibitor, 5 mM) for 3 days. The percentage of IgG1 B cells was analyzed by flow cytometry with four biological replicates ($n = 4$ biological replicates, $p < 0.001$) (**e**). The protein level of AID and H3K27Ac was quantified by western blotting ($n = 3$ biological replicates, * $p < 0.05$, ** $p < 0.01$, *** $p < 0.001$) (**f**). **g** Quantitative proteomic analysis of histone extracts from A20 cells cultured with U-$^{13}$C$_3$-pyruvate for 0, 6, 12 and 24 h with triplicate ($n = 3$ biological replicates). **h, i** Naive B cells isolated from *Mb1^Cre* (**h**) and *Mct1^{f/f}Mb1^Cre* (**i**) mice were cultured with Pyr or MePyr containing LPS and IL-4 for 3 days ($n = 3$ biological replicates, $p < 0.001$). The protein quantification analysis of H3K27Ac was determined by western blottingting. Data are presented as mean ± SD, unpaired two-tailed t-test (**a**–**d**, **f**, **h**, **i**), or two-way ANOVA followed by Sidak's multiple-comparisons test (**e**), *$p < 0.05$, **$p < 0.01$, ***$p < 0.001$, n.s not significant. Source data are provided as a Source Data file.

SLE is an autoimmune disease characterized by a series of autoantibodies against a variety of cellular components. Analyses of the clinical data suggest that *MCT1* expression is significantly increased in B cells from SLE patients. This upregulation is especially obvious in IgG class-switched memory B cells and activated naive B cells, mirroring the function of MCT1-mediated CSR in this report. These findings suggest a role for MCT1 in the potential pathogenesis and therapies of SLE. For example, the hyperactive MCT1 function in SLE patients may mediate the production of IgG autoantibodies for the development of SLE. Indeed, we showed that either *Mct1* deficiency or *Mct1* inhibitor treatment could significantly alleviate the levels of anti-dsDNA autoantibody in the BM12-induced lupus-like mouse model. The production of anti-dsDNA antibody may arise from either altered negative selection within GC or changed GC-derived IgG abundance. Although the exact mechanism needs further investigation, the data in this report support that the reduced anti-dsDNA IgG titers in B cell-specific *Mct1* deficiency mice are more likely a result of less bm12 cell induced IgG production as a consequence of reduced GC reactions, but less-likely a consequence of better negative selection of GCBC.

In summary, our study reveals a MCT1-dependent pyruvate-chromatin accessibility signaling transcriptional network that is essential for IgG antibody production in B cells. All these findings provide insight into the potential use of MCT1 inhibitors as a treatment for SLE.

## Methods

### SLE patient and healthy control subjects
PBMCs Samples were collected from a total of 24 SLE patients and 19 healthy controls by Peking University Affiliated People's Hospital for this study. HCs were age- and sex-matched individuals without autoimmune, inflammatory or infectious diseases. The protocol of evaluating human MCT1 transcription was approved by ethics committee for biomedical studies. All the volunteers involved in this current study have given their written consent. The ethics number is 2019PHB234-02.

### Chemicals and reagents
U-$^{13}$C$_6$-glucose (CLM-1396-PK) and U-$^{13}$C$_3$-pyruvate (CLM-1575-PK) were purchased from Cambridge Isotope. 2-DG was purchased from APExBio (B1027). LY294002 (S1105), AZD5363 (S8019), UK5099 (PZ0160), oligomycin (S1478), FCCP (S2876) and rotenone (S2348) were purchased from Sellck. Antimycin A was purchased from Santa Cruz (sc-202467). Pyruvate (P4562), lactate (71718), NAM (N0636), methylpyruvate (371173) and LPS (L2630) were purchased from Sigma. SRBC was purchased from Solarbio (S9702). IL-4 was purchased from Sino Biological (51084-MNAE). AZD3965 was purchased from Cayman (No. 19912). BMS-303141 (ACL inhibitor) was purchased from MCE (HY-16107).

### Mice and cell culture
C57BL/6J (JAX664) and bm12 (JAX1162) were purchased from the Jackson laboratory. *Mb1^Cre* mice were obtained from the Jackson

Laboratory (stock NO.020505). C57BL/6J (B6) background *Mct1^{f/f}* mice were designed with the CRISPR-Cas9 by the NIBS (National Institute of Biological Sciences, Beijing). Two floxed sites were inserted flanking *Mct1* exon 1 and intron 3, respectively (Supplementary Fig. 2a). The PCR identification primers, guide sequences of sgRNAs and donor sequences were shown in Supplementary Fig. 2b and Supplementary Data S1. Offsprings carrying *Mb1-Cre* and two copies of the floxed *Mct1* allele with 6-8 week were used in the experiments as homozygous mutant *Mct1^{f/f}Mb1^Cre* mice, and *Mb1^Cre* mice as control, respectively. Mice were maintained in separately ventilated cages in a specific pathogen-free (SPF) facility with 12/12-h light-dark cycle, in a room with temperature and light regulation, and the animals had unlimited access to food and water. All mice were maintained under specific pathogen-free conditions in the animal facility of Tsinghua University and used in accordance of governmental and institutional guidelines for animal welfare. All experimental studies were approved by the Institutional Animal Care and Use Committee (IACUC) of Tsinghua University. Primary B cells were isolated from the spleen of *Mct1^{f/f}Mb1^Cre* or *Mb1^Cre* mice as previously described[43]. The cells were cultured in RPMI1640 medium supplemented with 10% FBS, 50 μM β-mercaptoethanol, GlutaMax and nonessential amino acids. B cells were stimulated for 3 days using 10 μg/mL LPS and 50 ng/mL IL-4 to induce B cells CSR in vitro.

### Cell population classification
Bone marrow cells and splenocytes were disrupted and resuspended in PBS, then stained with different antibodies. T cells (CD19$^-$CD3e$^+$), B cells (CD19$^+$CD3e$^-$), Recirculating mature B cells (CD19$^+$IgD$^+$), Immature B cells (CD19$^+$IgD$^-$IgM$^+$CD43$^-$), Pro B cells (CD19$^+$IgD$^-$IgM$^-$CD43$^+$), Pre B cells (CD19$^+$IgD$^-$IgM$^-$CD43$^-$), Transitional B cells (CD19$^+$CD93$^+$), Follicular B cells (CD19$^+$CD93$^-$CD23$^+$CD21$^{low}$), Marginal B cells (CD19$^+$CD93$^-$CD23$^-$CD21$^{high}$), B1a (CD19$^+$CD23$^-$CD5$^+$CD43$^+$), B1b (CD19$^+$CD23$^-$CD5$^-$CD43$^-$) and B2 cells (CD19$^+$CD23$^+$) (Supplementary Fig. 14). GCBC (B220$^+$CD95$^+$GL7$^+$) and activated B cell (B220$^+$, CD95$^+$. CD86$^+$) (Supplementary Fig. 15).

### Immunization and ELISA assay
For GC analysis, 6-week-old *Mct1^{f/f}Mb1^Cre* mice and *Mb1^Cre* mice were injected intraperitoneally with $1 \times 10^9$ sheep red blood cells (SRBC, Solarbio, China), and analyzed at 7 days post-immunization. For T cell-dependent (TD) antigen NP$_{33}$-KLH (Bioresearch), 6-week-old *Mct1^{f/f}Mb1^Cre* mice and *Mb1^Cre* mice were injected intraperitoneally with 10 μg NP$_{33}$-KLH and analyzed at day 7 and 14 post immunization.

To detect NP-specific IgM, IgG, dsDNA-specific IgG and SmD-specific IgG in the 6-week-old *Mct1^{f/f}Mb1^Cre* mice and *Mb1^Cre* mice, 4 μg/mL NP$_{30}$-BSA or dsDNA or SmD were coated on 96 cells plates and incubated overnight at 4 °C. All these plates were blocked with 5% skimmed milk in PBS buffer at 37 °C for 2 h, followed by addition of the diluted mice serum into each well and incubation at 37 °C for

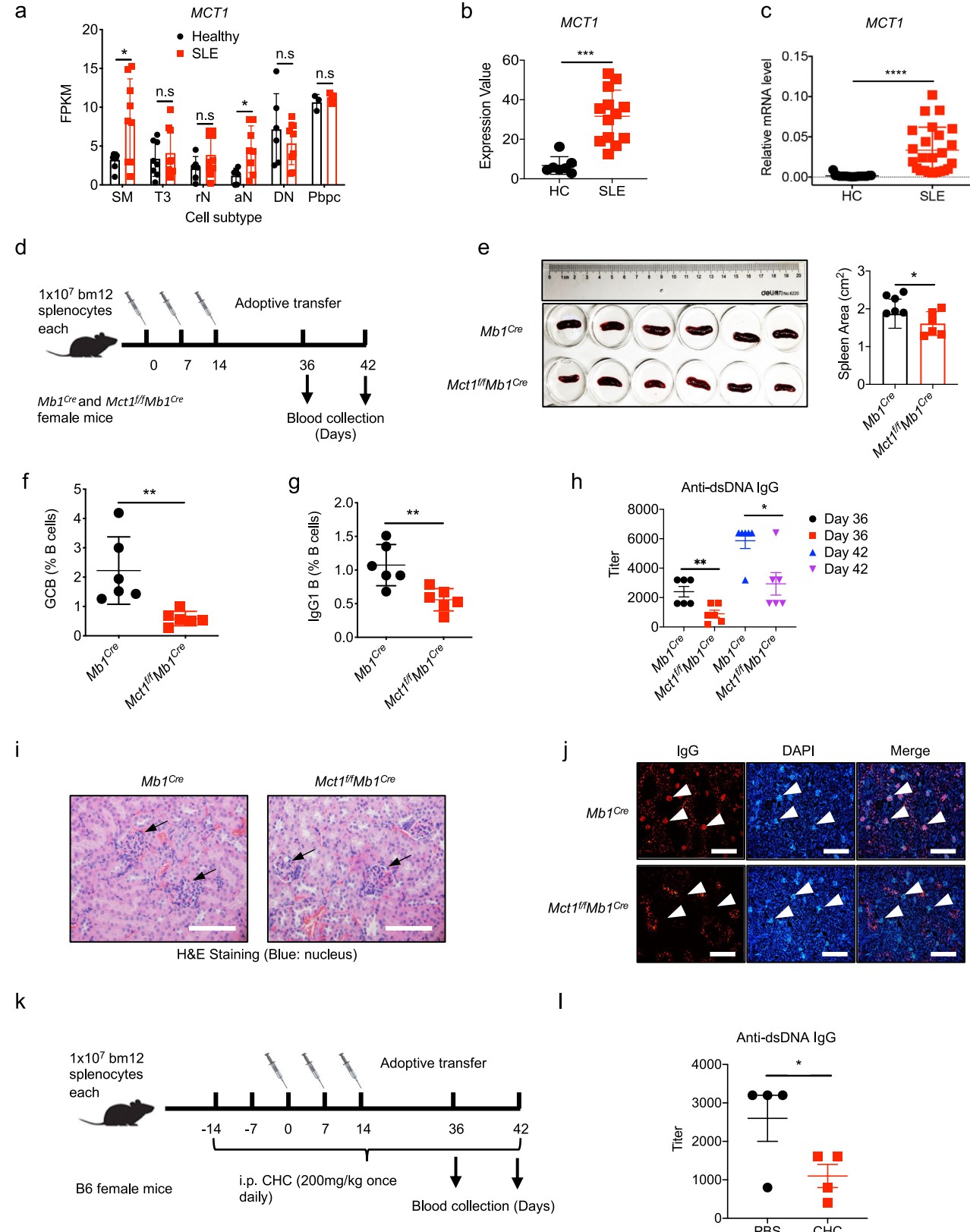

1 h. HRP conjugated goat anti-mouse IgM, IgG, and IgG3 (SouthernBiotech) were used. To determine the antibody titer, we measured the optical density (OD) of a series of dilutions of the primary antibody. The titer was defined as the highest dilution at which the OD was at least twice that of the blank control.

## Immunohistochemistry

Six-week-old *Mct1^f/f^Mb1^Cre^* mice and *Mb1^Cre^* mice were injected intraperitoneally with $1 \times 10^9$ sheep red blood cells (SRBC). The immunized or normal spleen sections were fixed in 4% paraformaldehyde-PBS solution at 4 °C overnight. The tissues were embedded in the paraffin,

**Fig. 6 | Mct1 deficiency in B cells suppresses IgG autoantibody production and GC formation in bm12-induced murine lupus model. a** *MCT1* expression in a cohort of African-American females (9 subjects with SLE and 12 healthy controls, from GEO accession: GSE118254). Activated Naive B cells (aN), Switched Memory B cells (SM), Resting Naive B cells (rN), Double negative B cells (DN), Transitional 3 B cells (T3), Antigen Secreting Cell (ASC) (Pbpc). **b** *MCT1* expression in B cells of 21 SLE patients and 18 healthy controls, from data of GSE10325 and GSE4588. **c** qPCR analysis of *MCT1* expression in healthy control (*n* = 19, *n* indicates different donors) and SLE samples (*n* = 24, *n* indicates different donors), *p* < 0.0001. **d**–**h** The schematic diagram of bm12-induced SLE model, *n* = 6 biological replicates (**d**). The spleen size of *Mb1^Cre^* and *Mct1^f/f^Mb1^Cre^* mice was calculated in bm12-induced SLE model (**e** *p* = 0.0105). The percentage of GCBC (**f** *p* = 0.0067) and IgG1 B cells

(**g** *p* = 0.0046) were analyzed in the bm12-induced SLE model. Anti-dsDNA IgG in *Mb1^Cre^* and *Mct1^f/f^Mb1^Cre^* mice serum was analyzed by ELISA (**h**), *p* = 0.0059(left), *p* = 0.0103(right). **i**–**k** H&E staining of kidney of *Mb1^Cre^* and *Mct1^f/f^Mb1^Cre^* mice. Black arrows indicated the inflammatory cell. Scale Bar, 25 μm (**i**). Immunofluorescence of IgG in the glomerulus of *Mb1^Cre^* and *Mct1^f/f^Mb1^Cre^* mice. White arrows indicated the glomerulus. Scale Bar, 100 μm (**k**). **l** The schematic diagram of CHC treatment procedure in SLE model. **m** The titer of anti-dsDNA IgG was determined by ELISA in WT or CHC-treated mice (*n* = 4 biological replicates, *p* = 0.0467). **d** and **k** were adapted from Biorender. Statistics (**e–k**, **m**) are representative one of three independent experiments. Data are presented as mean ± SEM, unpaired two-tailed t-test (**a**, **c**, **e**, **f**, **g**, **h**, **m**), *\*p* < 0.05, *\*\*p* < 0.01, *\*\*\*p* < 0.001, *\*\*\*\*p* < 0.0001, n.s. not significant.

or frozen in the OCT, and cut into 5 μm-thick sections, which were stained with H&E or Ki67 antibody (Abcam), FITC anti-mouse B220 antibody (Biolegend) and PNA (Yeasen, Shanghai, China), then observed with microscope (Olympus).

### MCT1 subcellular localization
The construct of MCT1 (NCBI accession NM_001166496.2) was also cloned into pMlink (GFP) plasmid for subcellular localization analysis using HEK293T cells. MCT1 localization was analyzed by Immunofluorescence in primary B cells using MCT1 antibody (Origene, #TA321556).

### Western blotting and Antibodies
The primary or cultured B cells were rinsed with PBS and lysed in RIPA (Beyotine, Shanghai, China) with complete protease inhibitor cocktail tablets (Roche, Mannheim, Germany). The following antibodies: anti-phospho-FOXO1 (Thr24) /FOXO3a (Thr32) (#9464), anti-phospho-AKT (Ser473) (#4060), anti-HK1 (#2024), anti-HK2 (#2867), anti-GAPDH (#2535), anti-PKM1/2 (#3106), anti-PKM2 (#4053), anti-LDHA (#3582), anti-PDHA (#3205), anti-H3K27Ac (#8173) and anti-H3 (#4499) were purchased from Cell Signaling Technology (CST). Anti-KI67 (#ab15580) and anti-Tomm20 (ab186735) were purchased from Abcam. Anti-LDHB (#14824-1-AP), anti-ALDOB (#18065-1-AP), Anti-MFN1 (#13798-1-AP), anti-MFN2 (#12186-1-AP), anti-OPA1 (#27733-1-AP), Pan (#66289-1-Ig) and anti-PC (#16588-1-AP) were purchased from Proteintech. Anti-β-actin (#30102ES40) was purchased from Yeasen. Anti-AID (#392500) was purchased from Thermo. Anti-MCT1 (#TA321556) was purchased from Origene. Cy™3 AffiniPure Fab Fragment Goat Anti-Mouse IgM, μ chain specific (#115-167-020), Alexa Fluor 488 Fab anti-IgM μ chain (#115-547-020) and Alexa Fluor 647 Fab anti-IgG1 Fcγ (#115-607-185) were purchased from Jackson ImmunoResearch Laboratories. Ef450 anti-GL-7 (#48-5902-80), ef660 anti-GL-7 (#50-5902-82), Alexa Fluor 488 anti-CXCR4 (#53-9991-80) and ef450 anti-IgM (#48-5790-82) were purchased from eBioscience. PerCP/Cyanine5.5 anti-B220 (#103209), FITC anti-CD95 (#152605), PE anti-CD95 (#152607), APC anti-CD86 (#105012), PE anti-CD3 (#100205), APC anti-CD19 (#115511), FITC anti-IgD (#405703), PE anti-CD43 (#143205), PE/Cyanine7 anti-CD93 (#136505), PE anti-CD23 (#101607), FITC anti-CD21 (#123407) and FITC anti-CD5 (#100605) were purchased from Biolegend. Biotin anti-CD19 (#553784) and anti-CD16/CD32 (#553140) were purchased from BD biology. Anti-κ chain (#1050-01), anti-IgM-HRP (#1020-05), anti-IgG-HRP (#1030-05), anti-IgG1-HRP (#1070-05), anti-IgG2b-HRP (#1090-05), anti-IgG2c-HRP (#1079-05) and anti-IgG3-HRP (#1100-05) were purchased from SouthernBiotech. All antibodies for western blotting are provided in Supplementary Data S2.

### Gene knockout of Mct1 in the A20 cells
The two targeting sgRNA sequences were constructed using a PX458-pSpCas9 (BB)−2A-GEP-MCS plasmid (a gift of Dr. Wei Guo) based on the protocol of Zhang Feng' s laboratory. Briefly, a pair of oligonucleotides is annealed, phosphorylated and ligated to a linearized

plasmid vector. Cells were sorted post-transfection in 96-well plates using a FACS Aria II cell sorter (BD BioSciences, San Jose, CA). Single-cell colonies were obtained and genomic DNA was extracted for further genotyping PCR and sequencing. The sgRNA sequences are as follows: 5'-CAACGACCAGTGAAGTATCA-3', 5'-TGATACTTCACTGGT CGTTG-3'.

### U-$^{13}$C$_6$-labled glucose metabonomics
Briefly, the primary B cells from the spleen of C57BL6J (B6) mice were treated for 30 min with U-$^{13}$C$_6$ glucose RPMI1640 medium containing LPS and IL-4 at day 0 to detect the metabolites by LC-MS/MS. The primary B cells from either *Mb1^Cre^* or *Mct1^f/f^Mb1^Cre^* mice were stimulated with cells culture medium containing the LPS and IL-4. At day 2, the culture medium was replaced with U-$^{13}$C$_6$ glucose (2 g/L), and both the cells and medium were harvested and detected at the time points of 30, 60 and 180 min upon the replacement. Metabolites were extracted from B cells and media with cold 80% methanol. Extracted metabolites were dried for LC-MS/MS analysis, which was performed as described previously[44]. Raw data are provided in Supplementary Data S4−5.

### Flow cytometry analyses of cell death, mitochondria mass, and transmembrane potential
Naive B cells or primary B cells treated with LPS and IL-4 for 3 days were stained with PI and Ki67 to detect the cell viability. The B cells treated with LPS and IL-4 for 3 days were stained with Mito-Tracker Green (Yeasen) for observing mitochondrial mass and Mito-Tracker CMXRos (Yeasen) to measure the mitochondrial transmembrane potential. All samples were analyzed by flow cytometry (BD BioSciences, San Jose, CA, USA).

### Glucose consumption and lactate production
Primary B cells were cultured with LPS and IL-4 for 3 days, after which the media was collected for glucose detection using the Glucose Consumption Assay kit (Applygen, Beijing). Intracellular or extracellular lactate and pyruvate were measured by the L-lactate and pyruvate assay kit (Elabscience) according to the manufacturer's protocol.

### Hexokinase (HK) activity
HK activity was assessed by using a HK activity assay kit (Solarbio, Beijing, China) following the manufacturer instructions.

### RNA extraction and qPCR analysis
Total RNA was extracted from B cells by using TRNzol reagent (Tiangen, China) and reverse transcribed by PrimeScript RT reagent kit (Takara). The RT-qPCR was performed using SYBR Green Master Mix (Transgen, Beijing, China) by an ABI ViiA™7 Real-Time System (Life Technologies). The primers (shown in Supplementary Data S1) were designed by the NCBI primer tools.

### Chromatin immunoprecipitation (ChIP) assays
For H3K27Ac ChIP assays, B cells from *Mb1^Cre^* and *Mct1^f/f^Mb1^Cre^* mice treated with LPS and IL-4 for 3 days were used. 2 × 10$^7$ cells were fixed

with 1% methanol-free formaldehyde for 10 min at room temperature. Fixation was halted by introduction of glycine for 5 min at room temperature. A ChIP assay was performed using the SimpleChIP enzymatic ChIP kit (Cell Signaling Technology). The bound DNA fragments were used for the ChIP-seq and RT-qPCR analysis. Primers used in this assay are listed in Supplementary Data S1.

## RNA seq and analysis

B cells from *Mb1$^{Cre}$* and *Mct1$^{f/f}$Mb1$^{Cre}$* mice were cultured with LPS and IL-4 for 3 days. RNA was extracted from whole cells with a RNeasy Plus Mini kit (Qiagen). Library data were assembled and analyzed by the BGISEQ institute. The upregulated and downregulated genes are listed in Supplementary Data S3. Affymetrix Microarray Data of GSE10325 and GSE4588 gene expression profile for B cells sorted from total PBMCs from both healthy individuals and SLE patients were download from the Gene Expression Omnibus (GEO) database. These two datasets included gene expression profiles for 21 SLE patients and 18 healthy controls. The matrix was based on GPL96 platforms (HG_U133A; Affymetrix Human Genome U133A Array) and GPL570 (HG_U133_Plus_2; Affymetrix Human Genome U133 Plus 2.0 Array). RNA-seq data of GSE118254 for six distinct B cell subsets isolated from a cohort of SLE and HC subjects was also download from GEO database. Differentially Expressed Genes (DEGs) between SLE patients and controls were identified using the LIMMA package downloaded from Bioconductor in R software (version 4.1.2). Analysis of high-throughput sequencing data revealed differential expression of *MCT1* in B cells from individuals with RA (GSE87095), MS (GSE190847) and SS (GSE146088), as well as changes in *MCT1* expression before and after drug administration in patients with SLE (GSE72798 and GSE159094).

Student's t-test in the LIMMA package was used to analyze significant difference between the groups. DEGs with adjusted $P < 0.05$ and |log2 fold change|>1 were considered to be statistically significant.

## ATAC-seq

ATAC-seq was performed using the aforementioned protocol[45]. Briefly, B cells from *Mb1$^{Cre}$* and *Mct1$^{f/f}$Mb1$^{Cre}$* mice were treated with LPS and IL-4 for 3 days in 6-well plates. Nuclei isolated from 50,000 B cells were used for ATAC-seq. B cells were harvested at $500 \times g$ 4 °C for 5 min, and then cell pellets were re-suspended in 100 μL of lysis buffer (10 mM Tris-HCl pH 7.4, 10 mM NaCl, 3 mM MgCl2, 0.5% NP-40) and then nuclei were centrifuged at $500 \times g$ 4 °C for 15 min. The nuclei pellets were re-suspended in the transposition reaction mix containing Tn5 transposase (Vazyme Biotech, Nanjing, China). Reactions were incubated for 30 min at 37 °C. The transposed DNA fragments were purified using the MinElute Kit (Qiagen) and the library was amplified using primers with unique barcodes (Vazyme Biotech, Nanjing, China), which were selected by the AMPureXP beads (Agencourt) to establish the library. The library was sequenced on an Illumina HiSeq X Ten sequencer for an average of 20 million unique reads per sample to obtain 150 bp paired-end reads.

## U-$^{13}$C$_3$-pyruvate labeling of histones

A20 cells were seeded into 10 cm dish and cultured with labeling medium (RPMI1640 glucose and pyruvate free medium supplemented with 2 mM U-$^{13}$C$_3$-pyruvate). At the indicated time points, cells were collected at $500 \times g$ for 5 min at 4 °C and rinsed with cold PBS containing 5 mM sodium butyrate, and suspended in the lysis buffer (15 mM TrisCl pH 7.5, 60 mM KCl, 15 mM NaCl, 5 mM MgCl$_2$, 1 mM CaCl$_2$, 250 mM Sucrose, 5 mM sodium butyrate, 1 mM DTT, 0.2% NP-40) with cOmplete protease inhibitor cocktail tablets (Roche, Mannheim, Germany) for 10 mins on ice. The nuclei were pelleted by centrifugation for 10 min at $500 \times g$ 4 °C, and washed twice with the lysis buffer lacking NP-40. Histones were extracted in 0.4 N H$_2$SO$_4$ at 4 °C with rotation for 4 h. The supernatants were collected at 3400 g

for 5 mins at 4 °C and analyzed by LC-MS/MS[46]. Raw data are provided in Supplementary Data S7.

## Hessian-SIM imaging

Cells Super-resolution imaging of mitochondrial morphology was performed using commercialized Hessian-SIM (High intelligent and Sensitive Microscope, HIS-SIM), provided by Guang zhou Computational Super-resolution Biotech Co., Ltd. Cells were cultured in 3.5 cm glass bottom dish pre-coated with Cell-talk (Corning) in RPMI1640 medium and maintained at 37 °C with 5% CO$_2$ in a humidified incubation chamber for live SIM imaging. For mitochondrial and nuclei morphology analysis, cells were activated with LPS and IL-4 for 2 days and then stained with Mitotracker Deep Red and Hoechst 33342 for 20 min at 37 °C. SIM images were randomly collected at least 15 fields in triplicate independent experiments and analyzed using Image J (v1.53t) with mitochondria analyzer[47].

## Electron microscopy (EM)

Cells were cultured with LPS and IL-4 in RPMI1640 medium for 2 days, and then fixed with 2.5% glutaraldehyde in 0.1 M PBS. Dehydration samples were embedded in Spurr epoxy resin and then cut into sections. Images were acquired with a EM (H-7650, Hitachi, Japan).

## Statistical analysis

All data were analyzed using the Prism 8 software (GraphPad). For in vivo assays, the data are shown as mean ± SEM. For in vitro experiments, data are shown as mean ± SD. Statistical analysis was performed using unpaired or paired Student's t-test and two-way ANOVA followed by Sidak's multiple-comparisons test (*$p < 0.05$, **$p < 0.01$, ***$p < 0.001$).

## Reporting summary

Further information on research design is available in the Nature Portfolio Reporting Summary linked to this article.

# Data availability

RNA Seq, CHIP-seq and ATAC-seq data generated by this study have been deposited in the Gene Expression Omnibus (GEO), under accession code GSE247204. *MCT1* expression in B cells of RA patients and healthy controls, were extracted from data of GSE87095. MCT1 expression in B cells of MS patients and healthy controls, were extracted from data of GSE190847. *MCT1* expression in B cells of SS patients and healthy controls, were extracted from data of GSE190847. *MCT1* expression in PBMCs upon conventional immunosuppressive drugs treatment of patients with lupus nephritis were extracted from data of GSE72798. *MCT1* expression in PBMCs with different types of in vitro treatment were extracted from data of GSE159094. The proteomics data are deposited at the ProteomeXchange Consortium via the PRIDE partner repository with the dataset identifier PXD047325. Source data are provided with this paper.

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

## Acknowledgements

We are grateful to the support of the Laboratory Animal Research Center of Tsinghua University. This work is supported by National Natural Science Foundation of China (92157301, 32130048, 31971085 and 91857108 to L.C.; 32141004 and 81825010 to W.L.; 82302036 to N.K.), the Ministry of Science and Technology of China National Key R&D Programs (2022YFA0806503 and 2018YFA0506903 to L.C.; 2021YFC2300500 and 2021YFC2302403 to W.L.), and Beijing Natural Science Foundation (23Z30090 to W.L.). This work is also supported by A Collaborative Fund from Cytocraft Biopharmaceutical Co., Ltd,

Tsinghua University Spring Breeze Fund (2021Z99CFY012 to L.C.; 2021Z99CFZ003 to W.L.),Tsinghua-Foshan Innovation Special Fund (TFISF, 2020THFS0133 to L.C.), Tsinghua Precision medicine foundation (No.2022TS013). We also thank the National Facility for Translational Medicine (Sichuan) for assistance with metabolic experiments.

## Author contributions

Conceptualization: L. Chen, W.L. Investigation: W.C., N.K., L.S., S.L., Y.L., L.T., Y.Z., B.W., R.C., X.C., L. Cheng, W.L., L. Chen. Methodology: W.C., S.L., N.K., L.S., Y.L., L.T., Y.Z., B.W., R.C., X.C., L. Cheng, X.L., H.D., W.L., L. Chen. Visualization: W.C., N.K., L.S., S.L., Y.L., C.Z., L.T., J.W., X.S., J.Y., Z.L., W.L., L. Chen. Writing, review and editing: Ligong Chen, W.L., W.C., N.K., L.S., S.L.,. Funding acquisition: L. Chen, W.L. and N.K.

## Competing interests

The authors declare no competing interests.
