## [Peer Review File · Nature Communications]

MCT1-governed pyruvate metabolism is essential for antibody class-switch recombination through H3K27 acetylationEditorial Note: This manuscript has been previously reviewed at another journal that is not operating a transparent peer review scheme. This document only contains reviewer comments and rebuttal letters for versions considered at *Nature Communications*.

REVIEWERS' COMMENTS

Reviewer #3 (Remarks to the Author):

I thank the authors for their efforts. The paper is much clearer now.

However, I still have one comment: In the present Figure 6H the anti dsDNA titers after bm12 cell oculation are depicted. Anti dsDNA antibodies represent autoantibodies arising from imperfect negative selection of (GC) B cells and are associated with Lupus. Changes in anti dsDNA antibody titers can result from altered negative selection or they can be linked to total IgG abundance, because many immune reactions produce autoreactive specificities. It is important to discriminate between these possibilities. The reduced anti dsDNA IgG titers in Mct1fl/fl x mb1Cre mice are likely either a result of less bm12 cell induced IgG as a consequence or reduced GC reactions; or they are a less-likely the consequence of better negative selection of B cells. These two possibilities should be acknowledged and discussed. As I proposed previously, normalizing anti dsDNA antibodies (by dividing the anti dsDNA titer by the total IgG titer in the very same samples, i.e. 36 and 42d after bm12 cell transfer) would be important here.

Reviewer #4 (Remarks to the Author):

This is a significant study. The authors have addressed all of the reviewer's concerns. But proofreading is necessary due to some grammar errors, such as "there are growing evidence (line 97)" and "Previous study (line 413)".

Reviewer #5 (Remarks to the Author):

The authors have addressed the comments of the three reviewers in an adequate manner.

In response to Reviewer #1, the authors clarified the discrepancy between their study and a previous study by Weisel et al. by focusing on in vitro-activated B cells. The authors also analyzed other metabolites transported by MCT1 and they found no effects of Mct1 deficiency on the intracellular abundance of lactate, acetoacetate and beta-hydroxybutyrate. The authors also explained the conflicting data in Figures 2 and 3 by different experimental designs.

In response to Reviewer#2, the authors quantified their immunoblots and indicated the number of independent experiments. They also examined the expression and H3K27 acetylation of genes encoding upstream regulators of AID expression and found no differences.

In response to Reviewer #3, the authors now provide a more detailed description of the data and the statistical analysis. They also quantified the immunoblots. As also criticized by Reviewer #1, the authors clarified the difference of the data in Figures 2 and 3 by the use of different experimental designs. Importantly, the authors addressed the concern of whether the results obtained in vitro are physiological. By using T-independent immunization, they observed a decrease of class-switched IgG3 in Mct1 mutant mice. The authors also examined the antibody affinity maturation by analyzing high vs. low affinity antibodies. In Mct1 mutant mice, they detect an approximately two-fold decrease in the abundance of high affinity antibodies. Finally, the authors corrected the mistakes in the labelling of figures.

Manuscript number: NCOMMS-23-33514A

Chi et al. “MCT1-governed pyruvate metabolism is essential for antibody class-switch recombination through the acetylation of H3K27”

Point-by-Point Reply to Editor’s and Reviewers’ Comments

We thank the reviewers and the editor for their critical evaluation of our manuscript. We find all suggestions insightful and constructive. As described below, we integrated our responses (in blue color font) to each comment into the revised manuscript by adding new data or editing the manuscript.

REVIEWERS' COMMENTS

Reviewer #3 (Remarks to the Author):

I thank the authors for their efforts. The paper is much clearer now.

However, I still have one comment: In the present Figure 6H the anti dsDNA titers after bm12 cell inoculation are depicted. Anti dsDNA antibodies represent autoantibodies arising from imperfect negative selection of (GC) B cells and are associated with Lupus. Changes in anti dsDNA antibody titers can result from altered negative selection or they can be linked to total IgG abundance, because many immune reactions produce autoreactive specificities. It is important to discriminate between these possibilities. The reduced anti dsDNA IgG titers in Mct1fl/fl x mb1Cre mice are likely either a result of less bm12 cell induced IgG as a consequence of reduced GC reactions; or they are a less-likely the consequence of better negative selection of B cells. These two possibilities should be acknowledged and discussed. As I proposed previously, normalizing anti dsDNA antibodies (by dividing the anti dsDNA titer by the total IgG titer in the very same samples, i.e. 36 and 42d after bm12 cell transfer) would be important here.

RESPONSE:

We appreciate this reviewer's constructive suggestion. Truly, the observed changes in anti dsDNA antibody titers in this report can “result from either altered negative selection within GC or they can be linked to total IgG

abundance". We fully took the suggestion from this reviewer that "These two possibilities should be acknowledged and discussed" when revising our manuscript. Still, we wish to perform a quick experiment to assess the contribution of these two potential models without mutually exclusive features. By examining the sera materials in our hands, we conducted an experiment using sera from WT mice with and without the treatment of MCT1 inhibitor, CHC, after bm12 cell transfer to measure the total IgG titer. The result from this new experiment (Supplementary Fig. 12f in the latest version of our manuscript) fully support the prediction from this reviewer that "The reduced anti dsDNA IgG titers in Mct1^{f/f}Mb1Cre mice are likely either a result of less bm12 cell induced IgG as a consequence of reduced GC reactions; or they are a less-likely the consequence of better negative selection of B cells.". Thus, we have made the necessary revisions and have added a relevant discussion for these two possibilities to the new version of our manuscript according to the suggestion from this reviewer that "These two possibilities should be acknowledged and discussed".

Reviewer #4 (Remarks to the Author):

This is a significant study. The authors have addressed all of the reviewer's concerns. But proofreading is necessary due to some grammar errors, such as "there are growing evidence (line 97)" and "Previous study (line 413)".

RESPONSE:

We would like to thank this reviewer for the insights helping us improve the quality of our manuscript. We also wish to apologize for these grammatical errors. We have thoroughly revised our manuscript and corrected the grammatical errors in the revised manuscript.

Reviewer #5 (Remarks to the Author):

The authors have addressed the comments of the three reviewers in an adequate manner. In response to Reviewer #1, the authors clarified the discrepancy between their study and

a previous study by Weisel et al. by focusing on in vitro-activated B cells. The authors also analyzed other metabolites transported by MCT1 and they found no effects of Mct1 deficiency on the intracellular abundance of lactate, acetoacetate and beta-hydroxybutyrate. The authors also explained the conflicting data in Figures 2 and 3 by different experimental designs.

In response to Reviewer#2, the authors quantified their immunoblots and indicated the number of independent experiments. They also examined the expression and H3K27 acetylation of genes encoding upstream regulators of AID expression and found no differences.

In response to Reviewer #3, the authors now provide a more detailed description of the data and the statistical analysis. They also quantified the immunoblots. As also criticized by Reviewer #1, the authors clarified the difference of the data in Figures 2 and 3 by the use of different experimental designs. Importantly, the authors addressed the concern of whether the results obtained in vitro are physiological. By using T-independent immunization, they observed a decrease of class-switched IgG3 in Mct1 mutant mice. The authors also examined the antibody affinity maturation by analyzing high vs. low affinity antibodies. In Mct1 mutant mice, they detect an approximately two-fold decrease in the abundance of high affinity antibodies. Finally, the authors corrected the mistakes in the labelling of figures.

RESPONSE:

We are very grateful for the supports and insights from this reviewer. All these are certainly crucial for us to improve the quality of our manuscript.